# Smoothelin-Like Protein 1 Regulates the Thyroid Hormone-Induced Homeostasis and Remodeling of C2C12 Cells via the Modulation of Myosin Phosphatase

**DOI:** 10.3390/ijms221910293

**Published:** 2021-09-24

**Authors:** Evelin Major, Ilka Keller, Dániel Horváth, István Tamás, Ferenc Erdődi, Beáta Lontay

**Affiliations:** Department of Medical Chemistry, Faculty of Medicine, University of Debrecen, H-4032 Debrecen, Hungary; evelinmajor@med.unideb.hu (E.M.); ilka.keller@med.unideb.hu (I.K.); horvath.daniel@med.unideb.hu (D.H.); tamas.istvan@med.unideb.hu (I.T.); erdodi@med.unideb.hu (F.E.)

**Keywords:** SMTNL1, thyroid hormone, C2C12 cell migration, myogenesis, myosin phosphatase, Na^+^/K^+^-ATPase

## Abstract

The pathological elevation of the active thyroid hormone (T_3_) level results in the manifestation of hyperthyroidism, which is associated with alterations in the differentiation and contractile function of skeletal muscle (SKM). Myosin phosphatase (MP) is a major cellular regulator that hydrolyzes the phosphoserine of phosphorylated myosin II light chain. MP consists of an MYPT1/2 regulatory and a protein phosphatase 1 catalytic subunit. Smoothelin-like protein 1 (SMTNL1) is known to inhibit MP by directly binding to MP as well as by suppressing the expression of MYPT1 at the transcriptional level. Supraphysiological vs. physiological concentration of T_3_ were applied on C2C12 myoblasts and differentiated myotubes in combination with the overexpression of SMTNL1 to assess the role and regulation of MP under these conditions. In non-differentiated myoblasts, MP included MYPT1 in the holoenzyme complex and its expression and activity was regulated by SMTNL1, affecting the phosphorylation level of MLC20 assessed using semi-quantitative Western blot analysis. SMTNL1 negatively influenced the migration and cytoskeletal remodeling of myoblasts measured by high content screening. In contrast, in myotubes, the expression of MYPT2 but not MYPT1 increased in a T_3_-dependent and SMTNL1-independent manner. T_3_ treatment combined with SMTNL1 overexpression impeded the activity of MP. In addition, MP interacted with Na^+^/K^+^-ATPase and dephosphorylated its inhibitory phosphorylation sites, identifying this protein as a novel MP substrate. These findings may help us gain a better understanding of myopathy, muscle weakness and the disorder of muscle regeneration in hyperthyroid patients.

## 1. Introduction

Thyroid hormones (THs) have important regulatory actions in skeletal muscle (SKM) [1,2,3]. The main forms of thyroid hormone produced and secreted by the thyroid gland are tetraiodothyronine (T_4_) and 3,3′,5-triiodothyronine (T_3_). Although it has been assumed that THs enter cells by passive diffusion due to their lipophilic properties, THs are primarily transported through the cell membrane by the monocarboxylate transporter 8 (MCT8) in SKM [4]. T_4_ is considered to be a prohormone and most of the peripheral effects of THs are ascribed to the action of T_3_. Intracellular monodeiodination of T_4_ by deiodinase type II (DIO2) produces T_3_, the active form of thyroid hormone in SKM [5]. The final level of thyroid hormone signaling control is given by the expression of nuclear thyroid hormone receptors (TRs), which positively or negatively regulate the transcription of TH responsive genes [3]. Functional TRs mediate the vast majority of TH actions on the level of gene transcription; however, so-called non-genomic effects using other target molecules have been reported as well [6].

Pathological elevation in serum or tissue TH levels results in the manifestation of hyperthyroidism, which is associated with significant alterations in differentiation, contractile function and the metabolism of SKM [3]. Above all, hyperthyroidism is often accompanied by muscle weakness in the face, throat and respiratory muscle, and wasting of muscles around the shoulders and hips leading to hyperthyroid myopathy [7,8]. Even though several cellular mechanisms have been identified in muscle tissues [9], which can explain the observed effects on the molecular level, the detailed molecular mechanism is not fully characterized yet.

Filamentous contractile motor protein myosin II consists of two 200 kDa heavy chains and two pairs of light chains of 17 and 20 kDa. Regulation of myosin II by phosphorylation is essential in muscle contraction or in the motile and other functions of non-muscle cells, e.g., shape changes, cell division, cell adhesion, cell migration or regulation of ion channels. After the phosphorylation of the 20 kDa myosin light chain (MLC20), myosin II reversibly binds to actin filaments via cross-bridges initiating contractile or motile events [10]. MLC20 phosphorylation depends on the balance between Ca^2+^/calmodulin-dependent myosin light chain kinase (MLCK) and myosin phosphatase (MP) [11].

Myosin phosphatase (MP) is a heterotrimer enzyme that hydrolyses phosphorylated myosin II light chains, as well as other cytoskeletal and cellular proteins such as adducin, tau, SNAP25 and PRMT5, etc. [12]. MP consists of a type 1 protein phosphatase catalytic subunit δ isoform (PP1cδ), a myosin phosphatase target subunit (MYPT) and a ~20 kDa small subunit of a yet unknown function (M20). The MYPT protein family includes MYPT1 (PPP1R12A) [13], MYPT2 (PPP1R12B) [14], MYPT3 [15], protein phosphatase 1 myosin binding 85 kDa subunit (MBS85) [16] and TGF-β1-inhibited membrane-associated protein (TIMAP) [17], which are responsible for targeting PP1cδ to its substrates [18]. MYPT1, first identified in smooth muscle, is ubiquitously expressed in most eukaryotic cells. MYPT2 was found mainly in cardiac and skeletal muscles [14,19].

The activity of MP holoenzyme is regulated through several mechanisms. The phosphorylation of MYPT by a variety of Ser/Thr kinases is one of the principal regulatory mechanisms leading to the activation or inhibition of MP. The phosphorylation of MYPT1 on Thr^696^ and/or on Thr^853^ results in the inhibition of MP activity [20,21]. In addition, the phosphorylation of Thr^853^ in MYPT1 by RhoA- associated kinase (ROCK) triggers the dissociation of MYPT1 from its substrate, myosin [22]. On the other hand, MP can be regulated via the formation of protein–protein interactions: an example is association with the smoothelin-like protein 1 (SMTNL1). SMTNL1 was identified in gastrointestinal smooth muscle as an early target of protein kinase A and G (PKA/PKG) [23,24]; however, it is also expressed in SKM and steroid hormone sensitive tissues. SMTNL1 inhibits the activity of MP, thereby regulating muscle contraction and cytoskeletal elements in various cells [25,26]. Through the inhibition of MP, SMTNL1 could enhance the phosphorylation of MLC20. Besides that, SMTNL1 is a transcriptional regulator of the progesterone receptor-B (PR-B), thereby regulating the gene expression of numerous metabolic enzymes, cytoskeletal proteins, steroid receptors and cytokines [26]. SMTNL1 is downregulated in hyperthyroid SKM and it selectively inhibits TRα expression, which is a key target of insulin-dependent signaling governed by T_3_. In addition, SMTNL1 overexpression reduced IRS1 Ser phosphorylation in a hyperthyroid model to restore the normal responsiveness of glucose transport to insulin [27].

Given the major role of thyroid hormones in the skeletal muscle system, it is important to study their actions as pathophysiological agents in myopathy. The objective of our work was to investigate the effect of supraphysiological vs. the physiological T_3_ level on the expression and regulation of MP via the modulation of SMTNL1 in myoblast differentiation and the homeostasis of C2C12 myoblasts and myotubes.

## 2. Results

### 2.1. Overexpression of SMTNL1 Promotes C2C12 Mouse Myoblast Differentiation

Myogenic differentiation is a highly coordinated process, which is essential for the development and regeneration of skeletal muscle. The mouse myoblast cell line C2C12 is a widely used model to study skeletal muscle myogenesis in vivo [28,29,30]. SMTNL1, a transcriptional cofactor, has been shown to regulate the adaptation of skeletal muscle to sexual development [26] and pregnancy [28] through altering the expression of contractile and metabolic proteins. In order to investigate the possible physiological relevance of SMTNL1 in skeletal muscle differentiation, SMTNL1 was transiently overexpressed in C2C12 cells. Following transfection, myoblasts were differentiated and Western blot analysis (Figure 1A,B) or immunofluorescence staining (Figure 1C) was performed. Desmin, a major intermediate filament protein, is present at low levels at early stages of myogenesis, and its expression increases as normal differentiation proceeds [29]. Our results demonstrate that desmin levels were significantly elevated in the NT-FT-SMTNL1-transfected cells compared to the empty vector-transfected control on Day three (Figure 1A). However, by the end of differentiation, the empty vector-transfected cells showed the same expression level of desmin as the NT-FT-SMTNL1-overexpressed cells. We also examined the expression of a late myogenic marker, myosin heavy chain (MyHC), the expression of which increases sharply during regular differentiation [30]. Here, we show that MyHC expression significantly rose on Day four and Day five due to SMTNL1 overexpression compared to the control (Figure 1B). To further confirm the upregulation of desmin expression caused by SMTNL1 overexpression, we performed immunofluorescence staining on empty vector-transfected or NT-FT-SMTNL1-transfected cells. As shown in Figure 1C, NT-FT-SMTNL1 overexpressing cells showed stronger fluorescent signals even at Day zero of differentiation, which was more pronounced on Day five compared to the control. These results demonstrate that SMTNL1 overexpression leads to the upregulation of early and late myogenic markers, suggesting that SMTNL1 promotes myoblast differentiation. To assess morphological changes throughout the differentiation of empty vector-transfected and NT-FT-SMTNL1-transfected cells, light microscopy images with the same parameter settings were taken and used to calculate the average area, average perimeter and myotube number using ImageJ software (Appendix A). We observed that SMTNL1 expression significantly decreases the average area and perimeter of myotubes on Day three and Day five of differentiation, respectively. However, SMTNL1 overexpression markedly increased the number of myotubes on Day three and Day five of differentiation. Taken together, these data indicate that SMTNL1 overexpression leads to accelerated myogenesis with myotubes that are significantly smaller in size but bigger in number.

### 2.2. Myosin Phosphatase Expression during C2C12 Myogenesis in the Presence of Excess T_3_

Intracellular T_3_ hormone availability depends on the presence of TH transporters in the plasma membrane, the activity of iodothyronine deiodinases and the expression of TRs, from which DIO2 and TRα were proved to play crucial roles in myogenesis, too [31,32]. Moreover, Wu et al. described the importance of myosin phosphorylation throughout the course of C2C12 differentiation, which is known to be partially dependent on the activity of MP [33]. Thus, we investigated the effect of excess T_3_ on these regulatory proteins during myoblast differentiation. MCT8 expression demonstrated a small increase during normal differentiation. In the presence of excess T_3_, MCT8 expression decreased slightly by Day six, which was 24% smaller compared to the vehicle control (Figure 2A). The expression of DIO2 demonstrated a sharp increase from Day three to Day four and continued to rise until the end of regular differentiation. In response to T_3_ treatment, DIO2 expression was induced consecutively until Day three and started to decline from Day four. By the end of differentiation, it was significantly diminished by 65% compared to the vehicle control (Figure 2B). In terms of TRα expression, it started to increase from Day four under regular differentiation. Upon T_3_ treatment, the TRα levels were significantly reduced from Day four until the end of differentiation compared to the vehicle control (Figure 2C). MYPT1/2 and MYPT2 expression was gradually increased during normal differentiation, while it showed an opposite tendency in the presence of excess T_3_ (Figure 3A,C). Despite the increase in MYPT1 expression on the first days of differentiation, it was reduced by Day six compared to the vehicle control (Figure 3B). Similarly, PP1cδ expression was moderately risen throughout normal differentiation; however, T_3_ overload caused an increase at first, and a modest decrease in its expression by the end of differentiation (Figure 3D).

### 2.3. SMTNL1 Overexpression Induces the Phosphorylation of MYPT1 and MLC20 in Myoblasts

SMTNL1 is known to inhibit the MP holoenzyme by directly binding to MP in the cytoplasm and by suppressing the expression of its regulatory subunit (MYPT1) at the transcriptional level [28]. Since MYPT1 is the main isoform in the MP present in non-differentiated C2C12 cells (Figure 3A,B) we examined the effect of SMTNL1 overexpression on the expression and phosphorylation of MP and its substrate, MLC20, under supraphysiological T_3_ exposure in myoblasts. Our data demonstrate that SMTNL1 expression was induced by 85% as a result of transfection in myoblasts (Appendix A). T_3_ treatment resulted in a slight decrease in the inhibitory phosphorylation of MYPT1, while the expression of MYPT1 showed no changes (Figure 4A,B). In contrast, SMTNL1 overexpression alone and in combination with T_3_ significantly reduced the expression of MYPT1, and enhanced the Thr696 phosphorylation of MYPT1 by 22 and 24%, respectively (Figure 4A,B). Remarkably, the SMTNL1-induced elevation in MYPT1 inhibitory phosphorylation was accompanied by an increase in the phosphorylation of MLC20 on Ser19 residue (Figure 4D). T_3_ treatment caused a small but not significant decrease in MLC20 phosphorylation (Figure 4D). The PP1cδ and MLC20 levels remained unchanged in response to any treatment (Figure 4C,E). Notably, SMTNL1 overexpression did not change the expression of MYPT2, while T_3_ treatment alone or in combination with SMTNL1 overexpression elevated MYPT2 expression (Appendix A). These results indicate that SMTNL1 has a major impact on MYPT1 in myoblasts.

### 2.4. SMTNL1 Overexpression Inhibits the Migration of Myoblasts

It has previously been shown that the silencing of MP results in the inhibition of migration [34]. Considering the significant impact that SMTNL1 has on the expression and phosphorylation level of MYPT1 and MLC20, we investigated the effect of SMTNL1 overexpression and T_3_ overload on the migratory capacity of C2C12 myoblasts since this ability is important in skeletal muscle differentiation and regeneration [35]. For this purpose, NT-FT-SMTNL1-transfected C2C12 cells labelled with fluorescent dye were uniformly scratched and cultured in the presence of supraphysiological T_3_ and changes were monitored for 24 h in real-time. Our results demonstrate that SMTNL1 overexpression alone or in combination with T_3_ markedly slowed down the closure of the scratch, while T_3_ treatment had negligible effects (Figure 5A). By monitoring the kinetics of cell migration, we detected a lower migration rate in the NT-FT-SMTNL1-transfected cells compared to the empty vector-transfected and to the T_3_-treated cells. Remarkably, this effect was more pronounced in response to the combined treatment with T_3_ and SMTNL1 overexpression (Figure 5B). As shown in Figure 5C, migratory capacity was slightly decreased upon T_3_ treatment, while SMTNL1 overexpression alone or in combination with T_3_ caused a 27 and a 34% decrease, respectively, compared to the empty vector-transfected control. Collectively, these data suggest that SMTNL1 negatively influences the migration of myoblasts, possibly through the inhibition of MP.

### 2.5. Differential Regulation of Myosin Phosphatase Target Subunit Isoforms by T_3_ Treatment and SMTNL1 Overexpression in Myotubes

Previous studies reported that there is a slight shift during myoblast differentiation from MYPT1 to MYPT2 [33]. Therefore, we examined these regulatory subunit isoforms of MP in differentiated C2C12 cells, too. Additionally, Lontay et al. reported that the expression levels of the MYPT2 isoform are unaffected by SMTNL1 deletion in developing muscle [26]. Thus, we investigated whether SMTNL1 overexpression affects MYPT2 expression or not. First, the overexpression of SMTNL1 was successfully confirmed using Western blot analysis showing a 43% increase in total SMTNL1 expression (Appendix A). Similarly to the results obtained on C2C12 myoblasts, the expression of MYPT1 did not change in response to excess T_3_ but it was significantly reduced by 23 and 38%, respectively, owing to SMTNL1 overexpression alone and in combination with T_3_ in myotubes (Figure 6A). MYPT2 expression was significantly increased by 56 and 64%, respectively, in response to excess T_3_ and to combined treatment with T_3_ and SMTNL1 overexpression, while it did not change upon SMTNL1 overexpression (Figure 6B). The phosphorylation of MYPT1/2 on Thr696 residue was significantly elevated upon T_3_ treatment by 29% and upon combined treatment with T_3_ and SMTNL1 overexpression by 53%, and it demonstrated a marginally significant increase due to the overexpression of SMTNL1 (Figure 6C). The phosphorylation of Thr853, another inhibitory phosphorylation site of MYPT1, was markedly enhanced under each condition compared to the empty vector-transfected control (Appendix A). Regarding PP1cδ, its levels were slightly increased following T_3_ treatment and/or SMTNL1 overexpression (Figure 6D). These results suggest that T_3_ and SMTNL1 are selective to the different MYPT isoforms.

### 2.6. Effect of T_3_ Treatment and/or SMTNL1 Overexpression on the Expression and Activity of the Na^+^/K^+^-ATPase, an Alternative MP Substrate in Myotubes

Since the expression and importance of the major substrate of MP, MLC20, decreases in SKM, we assumed that MP targets alternative substrate(s) in SKM. Moreover, we have identified Na^+^/K^+^-ATPase as a MYPT1 interacting protein using a pull-down assay followed by mass spectrometry analysis from a mouse synaptosome extract [36]. Thus, we examined the expression and activity of Na^+^/K^+^-ATPase in our current model system. Na^+^/K^+^-ATPase is a widely expressed transmembrane protein, which is essential to maintain whole body homeostasis as well as muscle function [37]. Its expression was found to increase in hyperthyroid muscle [38]. First, we studied the interaction between the Na^+^/K^+^-ATPase and MYPT in the C2C12 mouse myoblast cell line. Na^+^/K^+^-ATPase and MYPT were reciprocally co-precipitated. These data suggest that the Na^+^/K^+^-ATPase is capable of binding to MP, possibly through its regulatory subunit (Appendix A). The G-protein-coupled muscarinic receptor mimetic carbachol, as a positive control [39], and the PP1 selective inhibitor tautomycetin were applied on muscle strips, and the phosphorylation of Na^+^/K^+^-ATPase was assessed. Both the carbachol and TMC increased the phosphorylation of Ser23 in the α1 subunit of Na^+^/K^+^-ATPase (Figure 7A). We also investigated the effect of T_3_ treatment and/or SMTNL1 overexpression on Na^+^/K^+^-ATPase in C2C12 myotubes. T_3_ treatment alone and in combination with SMTNL1 overexpression significantly elevated Na^+^/K^+^-ATPase expression by 41 and 32%, respectively, while it remained unchanged upon SMTNL1 overexpression (Figure 7B). The phosphorylation of Na^+^/K^+^-ATPase on Ser16 and Ser23 residues was significantly increased in response to all the conditions (Figure 7C,D). These results indicate that both T_3_ and SMTNL1 regulates the activity of Na^+^/K^+^-ATPase by increasing its phosphorylation through the inhibition of distinct MYPT isoforms.

## 3. Discussion

Thyroid hormones are key regulators of the development, regeneration, metabolism and myogenesis of SKM [40]. Myogenic differentiation is a highly coordinated process, which is essential for the development and regeneration of SKM. During differentiation, myoblasts undergo remodeling and fuse to form multinucleated, contractile myotubes in parallel with an increased expression of muscle-specific proteins that coordinate the contractile and metabolic functions of SKM. Moreover, SMTNL1 expression is increased in SKM during development [26]. These data correlate with our findings that SMTNL1 overexpression moderately promoted the differentiation of C2C12 myoblasts to myotubes by increasing the levels of myogenic markers, such as MyHC and desmin (Figure 1). Moreover, SMTNL1 overexpression alters the phenotype of differentiated cells creating smaller myotubes (Appendix A).

In humans, the T_3_ level is maintained at low at the beginning of the myogenic process but it is increased parallel with the elevation of DIO2 upregulation [41]. It supports our data on C2C12 cell differentiation showing a significant increase in DIO2 expression upon T_3_ treatment; however, it markedly decreases in response to prolonged T_3_ treatment, suggesting a negative feedback mechanism (Figure 2B). The effect of MCT8 in myogenesis is debated since MCT8KO mice presented the same motor activity as WT mice [42]. In accord with that, we observed modest changes in MCT8 levels under each experimental setup during differentiation (Figure 2A). TRα was also changed during myoblast differentiation and showed a significant decrease upon prolonged T_3_ treatment (Figure 2C). TRα is crucial in the proliferation, migration and differentiation of myoblast cells and TRαKO mice showed impaired muscle regeneration [43]. In addition, SMTNL1, a co-regulator of TRα expression, is downregulated by supraphysiological T_3_ concentration and it hampers TRα expression [27]. Collectively, the C2C12 model proved to provide a sufficient hyperthyroid model for myogenesis with similar effects to that shown in human SKM. The effect of SMTNL1 in combination with the supraphysiological T_3_ level did not change the cell viability of differentiated C2C12 cells, suggesting that it has an impact not on the proliferation but the differentiation process itself [27].

The expression of MYPT isoforms and the enzyme activity of MP have already been assessed by Wu et al. [33] during C2C12 myoblast differentiation; they reported an increase in the expression level in MYPT2 and a decline in the MYPT1 isoform. These findings are supported by our data showing a slight increase in MYPT2 and a non-significant decline of MYPT1 expression (assessed by an anti-MYPT^1−38^ antibody that recognizes the MYPT1 isoform) [44] with no significant changes in PP1cδ expression (Figure 3). Interestingly, a supraphysiological T_3_ level elevated MYPT2 and PP1cδ expression but had no effect on MYPT1 expression, suggesting a selective regulation of MYPT isoforms. Surprisingly, the PP1-MYPT1 complex functioned even in differentiated C2C12 cells [33] and MYPT1 expression but not MYPT2 expression, was regulated by SMTNL1 (Figure 6A,B). Since MYPT1 and MYPT2 coding genes are located on chromosome 12q15–q21.2 and 1q32.1, respectively [14], their differential gene expression regulation is feasible. MYPT1 expression showed a significant decrease upon SMTNL1 overexpression, which was independent of T_3_-exposure in myoblasts and myotubes as well. It is in line with previous findings, showing that SMTNL1 downregulates the gene expression of MYPT1 in SKM [28]. MYPT2 expression is independent of SMTNL1 overexpression, which might be explained by the fact that SMTNL1 and MYPT1 are expressed exclusively in type 2A SKM fibers, while MYPT2 showed a homogenous distribution [28]. In the course of C2C12 differentiation, a shift from type 1 to 2A than to 2X and type 2B fiber marker proteins has been observed [45] and supraphysiological T_3_ exposure promotes this shift [46], supporting this hypothesis.

Non-muscle myosin (nm-myosin) has a fundamental role in processes that require cellular reshaping and movement, such as cell adhesion, cell migration, morphogenesis, development and cell division through its actin cross-linking and contractile functions in myoblasts [47]. The role of MP is inevitable in these processes [48]. Based on our data and other studies [33], MYPT1 is the major isoform in the MP present in non-differentiated C2C12 cells. Neither MYPT1 expression nor its activity was influenced significantly by T_3_ but only by the overexpression of SMTNL1 (Figure 4A,B). It has been reported that T_3_-induced ROCK activation resulted in an increased phosphorylation of MYPT1 in hepatic stellate cells [49]. It was also found to be time-dependent in osteoblast-like cells showing that the phosphorylation of MYPT1 was faded away by 6 h of T_3_ treatment [50]. It might explain the relatively low decrease in MYPT1 phosphorylation since our treatment lasted for 24 h (Figure 4B). In lung tissues, it was demonstrated that THs are able to modulate MLC20 phosphorylation and a moderate reduction in the level of phospho-MLC20 was detected in hyperthyroid mice compared to their euthyroid counterparts [51]. It supports our data on marginally decreased MLC20 phosphorylation upon T_3_-treatment, while it was greatly increased in response to the combined effect of T_3_ and SMTNL1 overexpression (Figure 4D). The T_3_ and SMTNL1-dependent phosphorylation of MLC20 resembles the migration activity and the ability of cytoskeletal remodeling of C2C12 cells. The combined effect of supraphysiological T_3_ and SMTNL1 hampered C2C12 migration; however, we need to note that drastic changes happened after 9 h of treatment (Figure 5B). The inhibition of MP by its phosphorylation at Thr696 could result in a slower turnover rate of nm-myosin phosphorylation causing more stable adhesive structures in the cells preventing migration and cytoskeletal remodeling. It is in accordance with a migration assay on HaCaT cells showing that PP1 enzyme inhibition started to cause a significant effect after 9 h of treatment [52]. In addition, SMTNL1 can also inhibit MP activity [26] by altering its phosphorylation that is clearly visible on MLC20 phosphorylation in SMTNL1-overexpressed myoblasts (Figure 4D). Since the MLC20 phosphorylation and MP activity were not fully in accordance with the migration features of the C2C12 myoblast, we assume that T_3_ has additional, most probably non-genomic effects on the cytoskeletal remodeling. These collectively lead to the arrest of myoblast migration and impede the remodeling of the actomyosin complex in non-differentiated C2C12 cells. We can conclude that MP facilitates myoblast migration that is hampered in a hyperthyroid state regulated by SMTNL1.

By the differentiation of C2C12 cells, a shift from the nm-myosin to the skeletal muscle type of myosin (sk-myosin) is observed, and at Day six the nm-myosin was barely detectable in the myotubes, while the concentration of sk-myosin was increased [33]. This is consistent with the slow rate of dephosphorylation of the sk-myosin phospho-light chain in vivo [53]. The MYPT1-PP1cδ smooth muscle-type holoenzyme shows low affinity to the phosphorylated sk-myosin and the sk-myosin phosphorylation was reduced but not eliminated at Day six of differentiation [33], suggesting that alternative substrates of myosin phosphatase exist in SKM.

One of the potential candidates as a substrate of MP is the Na^+^/K^+^-ATPase (NKA). NKA is a transmembrane pump that is essential for cellular ion and water homeostasis, maintaining membrane potentials and Na^+^-coupled transport of various substances. SKM has one of the largest and most dynamic pools of NKA in the body [37]. Therefore, the regulation of NKA either by the concentrations of its substrates or by protein–protein interactions with FXYD proteins or by covalent modification—predominantly by phosphorylation—is crucial [37]. We have already identified NKA as an MYPT1 interacting protein using a pull-down assay followed by mass spectrometry analysis from a mouse synaptosome extract [36]. The co-immunoprecipitation from C2C12 cell lysate also revealed the interaction between NKA and MYPT (Appendix A). In addition, NKA was proven to be a potential substrate of a PPP1 enzyme, possibly MP, due to the elevated phosphorylation at Ser23 residue upon the selective inhibition of PP1c by tautomycetin (Figure 7A). It is in line with previous studies showing that the inhibition of PP1c/PP2Ac by okadaic acid and calyculin A resulted in a decrease in NKA activity [54]. T_3_ stimulates the intrinsic activity of NKA in a transcription-independent manner via the stimulation of its cell surface expression through the PI3K/PKB pathway [55]. We have also detected this increase under the supraphysiological T_3_ exposure of C2C12 myotubes (Figure 7) and it is in accordance with findings showing the increased number of ouabain sites, also known as NKA level, in the SKM of hyperthyroid patients compared to healthy controls [56]. In addition, we proved that NKA gene expression occurs in an SMTNL1-independent manner (Figure 7B). Not only the protein level and translocation but the phosphorylation of the α subunit of NKA by Ser/Thr kinases can regulate the activity of NKA [55]. Although, the effects of THs are controversial due to the tight interplay between different signaling pathways; nevertheless, NKA phosphorylation by novel PKCs at Ser16 and Ser23 possessed mostly negative regulatory effect on NKA activity [57,58,59].

In myotubes, the expression of MYPT2 but not MYPT1 increased in a T_3_-dependent but SMTNL1-independent manner along with a marginal increase in the expression of PP1 catalytic subunit. Interestingly, T_3_ and its combined treatment with SMTNL1 overexpression impeded the activity of MP assessed by its regulatory Thr696 phosphorylation (Figure 6C). Parallel with these processes, the phosphorylation of NKA at Ser16 and Ser23 also significantly increased upon T_3_ treatment and SMTNL1 overexpression as well (Figure 7C,D). We strongly suggest that SMTNL1 took effect on increased NKA phosphorylation and activity via the downregulation of MP. The phosphorylation and inactivation of MP might be evoked by ROCK that was stimulated by T_3_ in osteoblasts resulting in the phosphorylation of MYPT1 at Thr696 and hampered osteocalcin synthesis [50]. It is important to note that both the overstimulation of NKA expression by T_3_ or hyper- or downregulation of its activity do not necessarily translate into improved muscle performance, and it could cause muscle weakness and myopathy [37].

In conclusion, we suggest a differential regulation and role of MP in myoblasts and myotubes upon supraphysiological T_3_ exposure. In non-differentiated myoblasts, MP is in complex with MYPT1 and its expression and activity is regulated by SMTNL1 that modulates the MLC20 phosphorylation, cytoskeletal remodeling and migration. In contrast, in myotubes, where the action of MP is less profound in the regulation of muscle contractility, since sk-myosins are regulated by a separate set of proteins that are bound to the actin filaments, and in the MP holoenzyme, PP1c is predominantly complexed with MYPT2. MYPT2 expression is stimulated by T_3_, and MP is regulated by the T_3_-induced ROCK phosphorylation. MP acts as a possible regulator of NKA by dephosphorylating its inhibitory phosphorylation sites. These findings should lead to a better understanding of myopathy and muscle weakness and the disorder of muscle regeneration in hyperthyroid patients.

## 4. Materials and Methods

### 4.1. Chemicals

All chemicals were obtained from Sigma-Aldrich (St. Louis, MO, USA) unless otherwise indicated.

### 4.2. Antibodies

Western blot analyses were performed using antibodies specific for desmin 1:1500 (Cell Signaling Technology, Danvers, MA, USA, #5332), MyHC 1:1500 (R&D Systems, Minneapolis, MN, USA, MAB4470), Flag 1:2000 (Sigma-Aldrich, St. Louis, MO, USA, F7405), MYPT^1−296^ 1:1000 [44], MYPT^1−38^ 1:1000 [33], MYPT1 1:1000 (BD Transduction Technologies, Becton, NJ, USA, #612164), PP1cδ 1:1000 (Santa Cruz Biotechnology, Dallas, TX, USA, #sc-365678), MYPT1-P^T696^ 1:1000 (Merck Millipore, Burlington, MA, USA, #ABS45), MYPT1-P^T853^ 1:500 (Merck Millipore, #36-003), MLC20 1:1000 (Cell Signaling Technology, #3672), MLC20-P^S19^ 1:500 (Cell Signaling Technology, #3671), MYPT2 1:1000 [14], Na^+^/K^+^-ATPase 1:1000 (Cell Signaling Technology, #3010), Na^+^/K^+^-ATPase-P^S16^ 1:250 (Cell Signaling Technology, #4020), Na^+^/K^+^-ATPase-P^S23^ 1:500 (Cell Signaling Technology, #4006), MCT8 1:1500 (Proteintech, Rosemont, IL, USA, #20676-1-AP), DIO2 1:1500 (NovusBio, Littleton, CO, USA, #NBP1-00178), TRα 1:650 (Abcam, Cambridge, UK, ab53729), SMTNL1 (Santa Cruz Biotechnology, #sc-390369), β-actin 1:20,000 (Santa Cruz Biotechnology, #sc-47778), α-actinin 1:8000 (Sigma-Aldrich, A2543) α-tubulin 1:1000 (Santa Cruz Biotechnology, #sc-53646), anti-mouse 1:5000 (Cell Signaling Technology, #7076), anti-rabbit 1:5000 (Sigma-Aldrich, A0545), anti-goat 1:5000 (Sigma-Aldrich, A5420).

### 4.3. Cell Culture Maintenance

C2C12 mouse myoblast cells (ECACC 91031101) were maintained in Dulbecco’s Modified Eagle’s Medium (DMEM) containing 1000 mg/L glucose, 2 mM L-glutamine (Lonza; Basel, Switzerland), 10% (*v*/*v*) foetal bovine serum (FBS) and phenol red. Cells were grown in a humidified incubator that provided an atmosphere of 5% CO_2_ at 37 °C.

### 4.4. Thin Coating of Tissue Culture Plates

Rat tail collagen I (Thermo Fisher Scientific; Waltham, MA, USA) was diluted in 20 mM acetic acid at a final amount of 5 µg/cm^2^ and was added to each well. After 1 h, the solution was removed, and wells were rinsed twice with 1× phosphate buffered saline (PBS) to remove the acid. Plates (VWR International; West Chester, PA, USA) were used immediately or air dried and stored at 4 °C until use.

### 4.5. Transient Transfection and Differentiation

GeneJuice transfection reagent was added dropwise to serum-free DMEM and incubated at room temperature (RT) for 5 min. According to the manufacturer’s intructions, 2 µg of pM13-NT-FT-SMTNL1 per well was added to the serum-free DMEM/transfection reagent mixture and was incubated at RT for 15 min. Meanwhile, cells were washed with 1× PBS and were detached with 0.25% (*w*/*v*) trypsin/EDTA solution. After the 15-min incubation ended, the entire volume of the transfection mixture was added dropwise to cells in complete growth medium. Finally, cells were seeded in 6-well tissue culture plates and incubated at 37 °C overnight (O/N). Transfection with empty vector (MOCK) acted as the control in all experiments. The day after transfection, cells were washed with 1× PBS and complete growth medium was replaced by phenol red-free DMEM supplemented with 1000 mg/L glucose, 2 mM L-glutamine and 2% (*v*/*v*) horse serum (HS). From then on, differentiation medium was changed daily for 5 days.

### 4.6. Immunofluorescence Staining

Transfected myoblasts were seeded into collagen-coated OptiPlate-96 Black plates (PerkinElmer; Waltham, MA, USA) and were differentiated for 5 days. On days 0, 3 and 5 of differentiation, cells were fixed with 4% (*w*/*v*) paraformaldehyde for 15 min at RT. Then, cells were washed 3 times with 1× PBS and were permeabilized with 0.2% (*w*/*v*) Triton-X-100 for 4 min at RT. After blocking with 1% HS for 1 h at 4 °C, immunofluorescence staining was carried out using anti-desmin (1:100) antibody (green), anti-Flag (1:100) antibody (green) and F-actin-binding peptide phalloidin (1:500) (red). Fluorescent signals were detected using an automated high-content imaging reader (PerkinElmer; Waltham, MA, USA).

### 4.7. 3,3′,5-Triiodo-L-thyronine (T_3_) Hormone Treatment

According to the manufacturer’s instructions, 3,3′,5-Triiodo-L-thyronine sodium salt was dissolved in 1 M HCl:EtOH with a ratio of 1:4. In Figure 2 and Figure 3, cells received 10 nM of T_3_ from Day 1 to Day 6 of differentiation. In Figure 4, 48 h after transfection, myoblasts were treated with 10 nM T_3_ for 24 h. In Figure 6 and Figure 7, myoblasts were transfected the day before the beginning of differentiation. From the beginning of Day 4 to the end of Day 6 of differentiation, differentiation medium was supplemented with 10 nM of T_3_. In all cases, equal amounts of HCl:EtOH (1:4) were added to the non-treated control called ‘vehicle’ in Figure 2 and Figure 3.

### 4.8. Cell Lysis

Myotubes were washed with 1× PBS and were harvested in ice-cold RIPA buffer complemented with protease and phosphatase inhibitors (25 mM Tris-HCl (pH 7.6), 150 mM NaCl, 1% sodium deoxycholate, 0.1% SDS, 1% Triton-X 100, 1 mM PMSF, 10× protease inhibitor cocktail, 50× phosphatase inhibitor cocktail, 1 µM microcystin-LR). Cell suspensions were sonicated for 30 s with a 10% pulse using a Branson Sonifier 250 sonicator (Thermo Fisher Scientific; Waltham, MA, USA) and were centrifuged at 16,000× *g*, 4 °C for 10 min. Supernatants were transferred to fresh tubes and total protein concentration was measured with a Bicinchoninic acid (BCA) assay kit (Thermo Fisher Scientific; Waltham, MA, USA).

### 4.9. Western Blot Analysis

Whole-cell lysates were boiled in 4× sodium dodecyl sulphate (SDS) sample buffer (Bio-Rad Laboratories; Hercules, CA, USA) at 100 °C for 5 min. A total of 30 µg of protein was loaded onto 4–20% precast Criterion gels (Bio-Rad Laboratories; Hercules, CA, USA) and was separated by size at a constant voltage of 200 V. Next, proteins were transferred onto nitrocellulose membranes using a Criterion blotter (Bio-Rad Laboratories; Hercules, CA, USA) at 100 V for 75 min. Membranes were blocked in 5% (*w*/*v*) bovine serum albumin (BSA) dissolved in 1× Tris buffered saline (TBS) containing 0.1% (*v*/*v*) Tween 20 at RT for 1.5 h. After that, membranes were incubated with primary antibodies, diluted in 5% BSA/TBST, at 4 °C O/N. On the next day, membranes were incubated with HRP-conjugated secondary antibodies at RT for 1.5 h and immunoreactions were visualized by enhanced chemiluminescence (ECL) using a mixture of WesternBright ECL or WesternBright Sirius and WesternBright Peroxide reagents (Advansta Inc.; San Jose, CA, USA) with a ratio of 1:1 in a ChemiDoc Touch Imaging System (Bio-Rad Laboratories; Hercules, CA, USA). In case of phospho-blots, membranes were stripped as described in Appendix A and were reprobed with non-phospho antibodies.

### 4.10. Scratch Assay and High Content Screening (HCS) Analysis

Myoblasts were suspended in serum-free medium containing 5 µL/mL of DiI (Thermo Fischer Scientific; Waltham, MA, USA), a highly lipophilic carbocyanine dye, and were incubated for 15 min at 37 °C. After washing 3 times with warm complete medium, myoblasts were transfected with an empty vector or NT-FT-SMTNL1 and were seeded into collagen-coated OptiPlate-96 Black plates (PerkinElmer; Waltham, MA, USA). After 48 h, cells were serum-starved for 5 h and were scratched with a 10-microliter pipette tip. Myoblasts were washed with 1× PBS and were cultured in the presence of 10 nM T_3_ for an additional 24 h under real-time screening with an HCS equipment using a 10× air objective. Data were analyzed and evaluated using the Harmony software (PerkinElmer; Waltham, MA, USA).

### 4.11. Carbachol and Tautomycetin Treatment

Mice were housed as described before [28]. All procedures were approved by the University of Debrecen Faculty of Medicine Ethical Committee and are consistent with the NIH ‘Guide for the Care and Use of Laboratory Animals’. Plantaris muscle strips were removed and maintained as described before [24] in an organ bath filled with Krebs solution (118 mM NaCl; 4.75 mM KCl; 1.2 mM MgSO_4_; 1.2 mM KH_2_PO_4_; 2.5 mM CaCl_2_; 25 mM NaHCO_2_; 11.5 mM glucose) warmed at 37 °C and gassed with 95% O_2_ and 5% CO_2_. The effects of PP1 inhibitor on muscle tissues were investigated by a 30-min preincubation with 10 μM tautomycetin (TMC) followed by a 10-min incubation with 10 μM carbachol. Tissues were lysed and subjected to Western blot analysis.

### 4.12. Statistical Analysis

Immunoblots were analyzed using ImageJ software. Bar charts were created and statistical analyses were performed using GraphPad Prism 8 software. Phosphorylated proteins were normalized to non-phosphorylated protein expression, while non-phosphorylated protein expression was normalized to the loading control. All data are presented as mean ± SEM, where *n* is the number of independently performed experiments. Differences were considered to be statistically significant at *p* < 0.05.

## Figures and Tables

**Figure 1 ijms-22-10293-f001:**
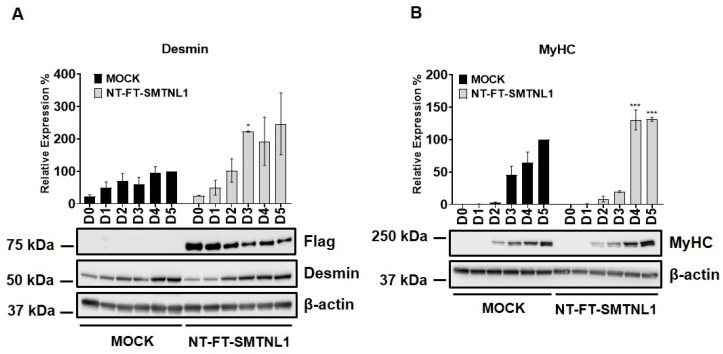
Effect of SMTNL1 overexpression on desmin and MyHC expression during myogenesis. Lysates from NT-FT-SMTNL1 or empty vector-transfected differentiated C2C12 cells were analyzed using Western blot with antibodies specific for early and late differentiation markers desmin (**A**) and MyHC (**B**), respectively. Transfection efficiency was determined using immunoblotting with anti-Flag antibody. Values represent *n* = 3, mean ± SEM. Differences between group means were determined using Two-way ANOVA followed by Sidak multiple comparisons post hoc test, *p* < 0.05 (*), *p* < 0.001 (***). (**C**) Immunofluorescent staining of cells at Day 0, Day 3 and Day 5 of differentiation after transfection with empty vector or NT-FT-SMTNL1. For staining, anti-desmin antibody (green) and Texas-Red Phalloidin (red) were used. Fluorescent signals were detected using an automated high-content screening instrument. Scale bars: 500 µm.

**Figure 2 ijms-22-10293-f002:**
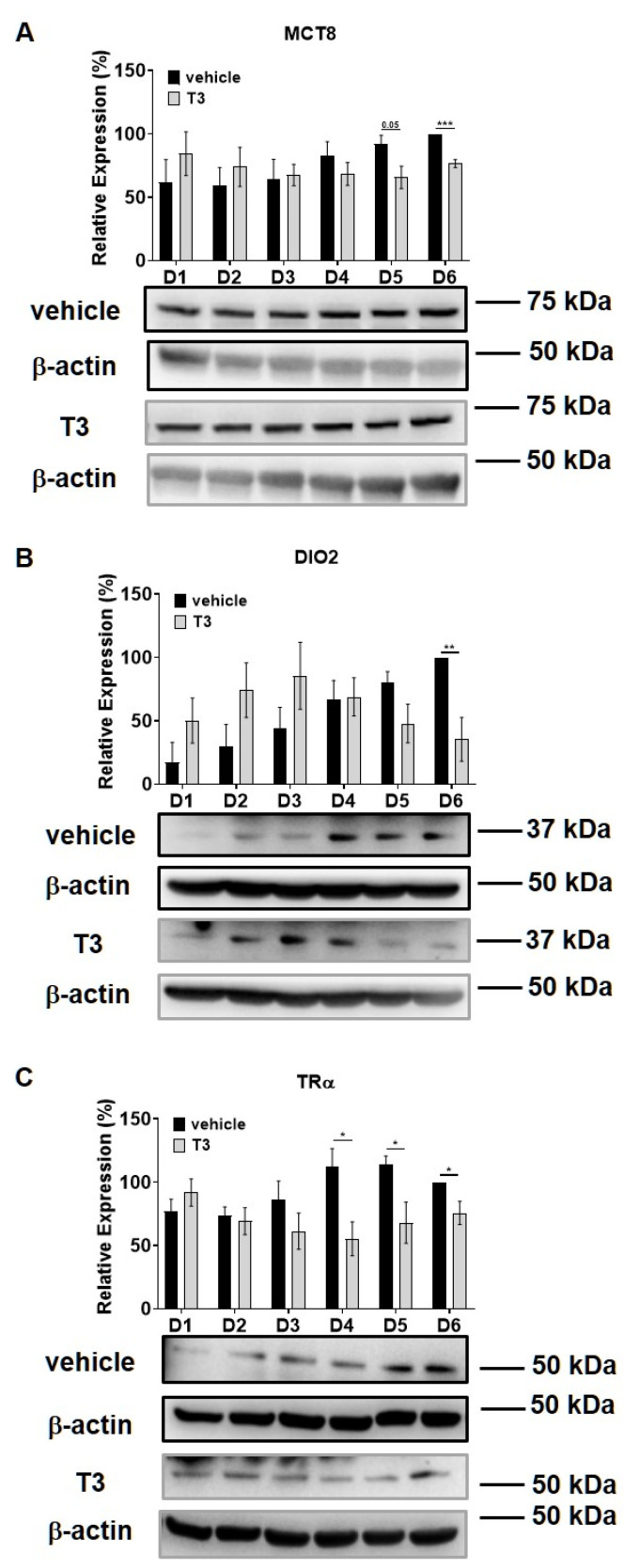
MCT8, DIO2 and TRα expression throughout the differentiation of C2C12 myoblasts in the presence of T_3._ (**A**–**C**) Myoblasts were differentiated for 6 days receiving either 10 nM T_3_ or equal amounts of HCl:EtOH solution. Proteins from whole-cell lysates were analyzed using Western blot with anti-MCT8 (**A**), anti-DIO2 (**B**) and anti-TRα (**C**) antibodies. Values represent *n* = 4–5, mean ± SEM. Data were normalized to the Day 6 vehicle control. Differences between group means were determined using Two-way ANOVA followed by Sidak post hoc test and by multiple unpaired two-tailed *t*-tests, *p* < 0.05 (*), *p* < 0.01 (**), *p* < 0.001 (***).

**Figure 3 ijms-22-10293-f003:**
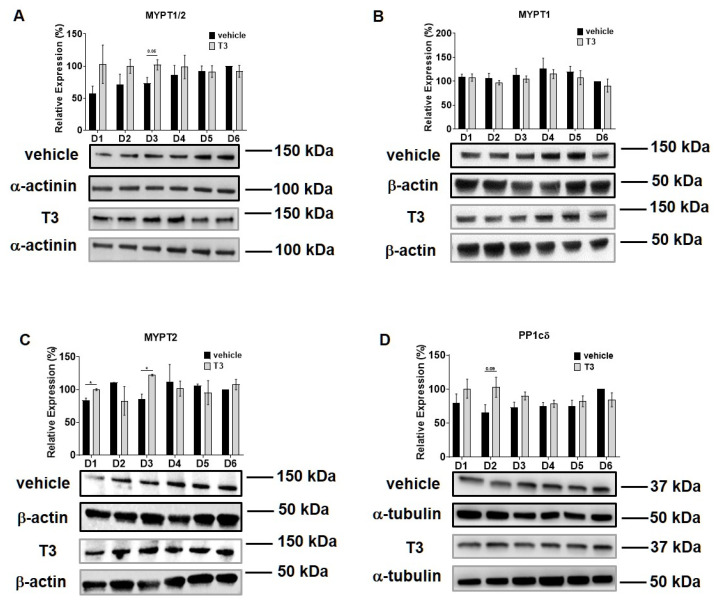
MYPT1/2 and PP1cδ expression during C2C12 myogenesis in the presence of T_3_. Myoblasts were differentiated for 6 days in the presence of 10 nM T_3_ or equal amounts of HCl:EtOH solution. Proteins from whole-cell lysates were analyzed using Western blot with anti-MYPT^1−296^ (**A**), anti-MYPT^1−38^ (**B**), anti-MYPT2 (**C**) and anti-PP1cδ (**D**) antibodies. Values represent *n* = 4, mean ± SEM. Data were normalized to the Day 6 vehicle control. Differences between group means were determined using Two-way ANOVA followed by Sidak post hoc test and by multiple unpaired two-tailed *t*-tests, *p* < 0.05 (*).

**Figure 4 ijms-22-10293-f004:**
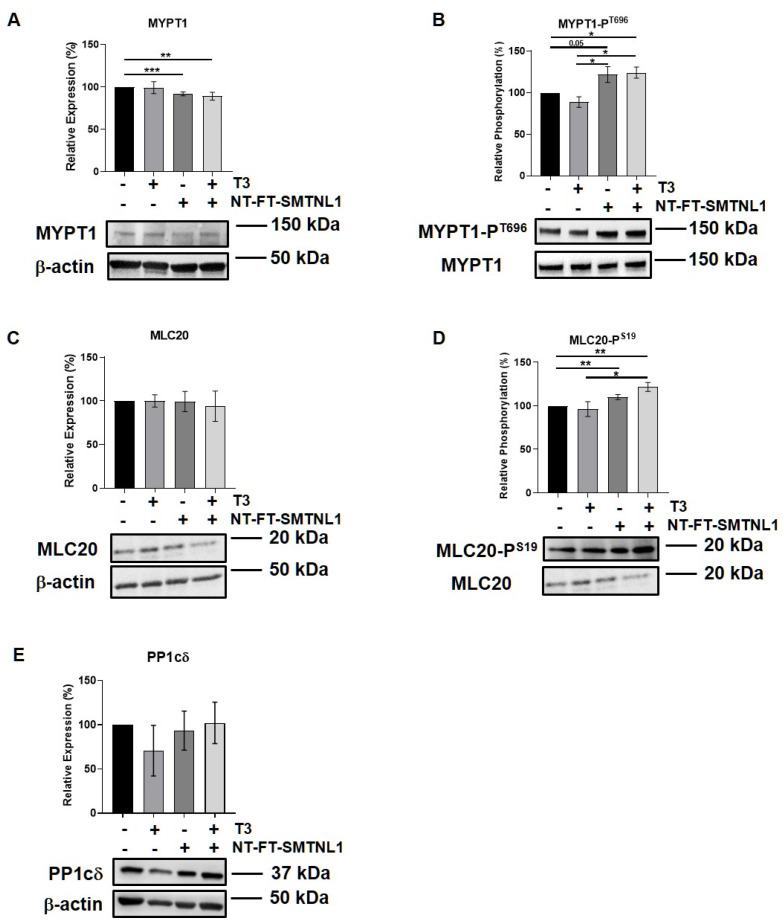
SMTNL1 overexpression controls the phosphorylation of myosin phosphatase and its substrate in myoblasts. Myoblasts were transfected with empty vector or NT-FT-SMTNL1 and were treated with 10 nM T_3_ for 24 h. Whole-cell lysates were analyzed using Western blot with anti-MYPT1 (**A**), anti-MYPT1-P^T696^ (**B**), anti-MLC20 (**C**), anti-MLC20-P^S19^ (**D**) and anti-PP1cδ (**E**) antibodies. Values represent *n* = 3, mean ± SEM. Data were normalized to the empty vector-transfected control. Differences between group means were determined using One-way ANOVA and Tukey’s post hoc test, *p* < 0.05 (*), *p* < 0.01 (**), *p* < 0.001 (***).

**Figure 5 ijms-22-10293-f005:**
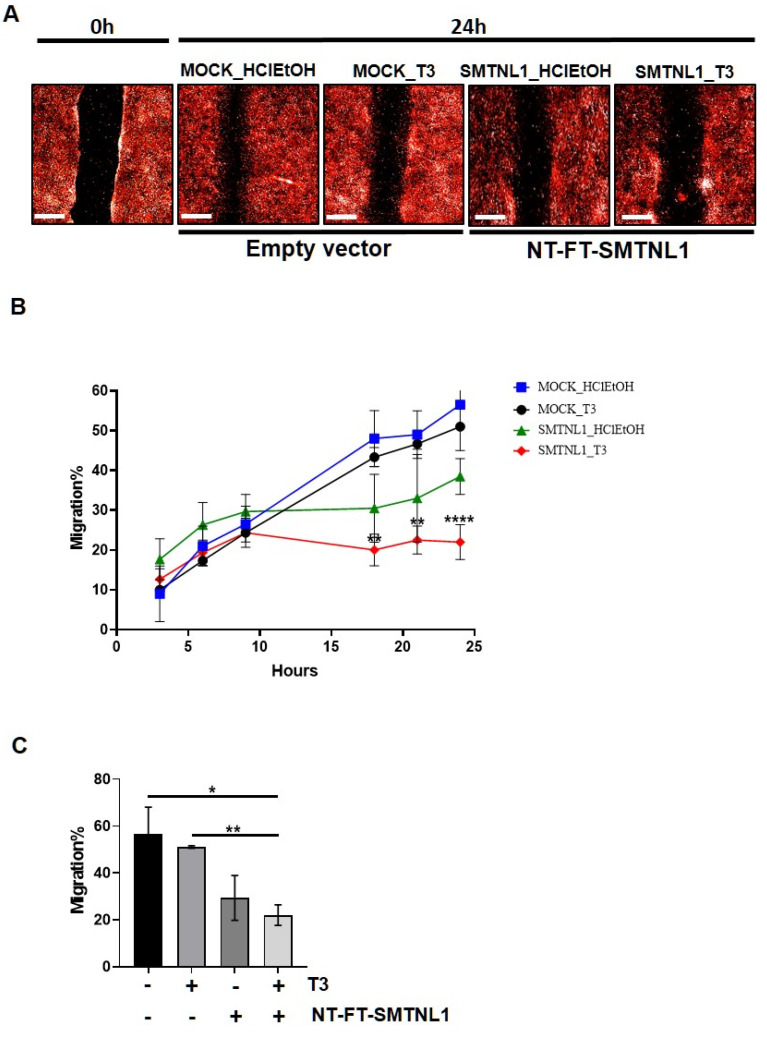
Scratch assay of T_3_-treated and SMTNL1 overexpressing myoblasts. Myoblasts were stained and transfected with empty vector or NT-FT-SMTNL1. After 48 h of culturing and a 5-h serum starvation, cells were scratched and were grown for additional 24 h in the presence of 10 nM T_3_. (**A**) Photos of scratch areas were taken with a 10× air objective at the indicated times. Scale bars: 500 µm. (**B**) 24-h kinetic curve of migrating C2C12 myoblasts. (**C**) Bar charts show covered scratch areas after 24 h. Values represent *n* = 3, mean ± SEM. Data were normalized to the empty vector-transfected control (MOCK). Differences between group means were determined using Two-way ANOVA (**B**) or One-way ANOVA (**C**) followed by Tukey’s post hoc test, *p* < 0.05 (*), *p* < 0.01 (**), *p* < 0.0001 (****).

**Figure 6 ijms-22-10293-f006:**
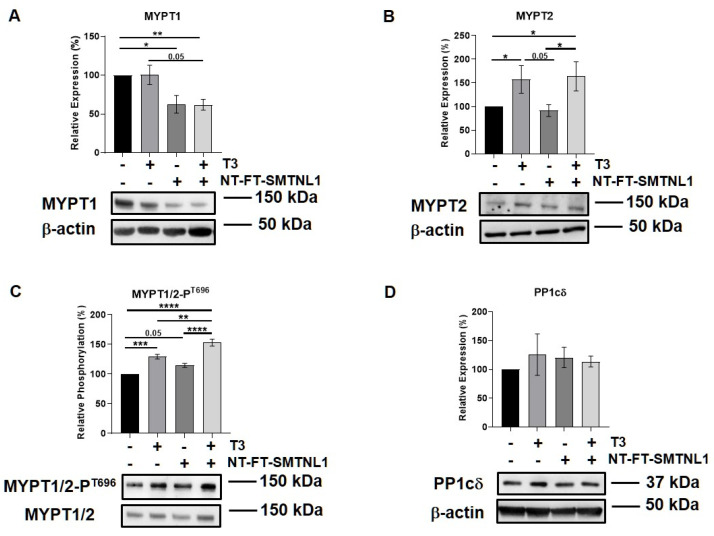
T_3_ Treatment and SMTNL1 overexpression influence the expression and phosphorylation of myosin phosphatase target subunits in myotubes. Myoblasts were transfected with empty vector or NT-FT-SMTNL1 and were differentiated with a simultaneous 72-h T_3_ treatment started from Day 4. Whole-cell lysates were analyzed using Western blot with anti-MYPT1 (**A**), anti-MYPT2 (**B**), anti-MYPT1/2-P^T696^ (**C**) and anti-PP1cδ (**D**). Values represent *n* = 3, mean ± SEM. Data were normalized to the empty vector-transfected control. Differences between group means were determined using One-way ANOVA and Tukey’s post hoc test, *p* < 0.05 (*), *p* < 0.01 (**), *p* < 0.001 (***), *p* < 0.0001 (****).

**Figure 7 ijms-22-10293-f007:**
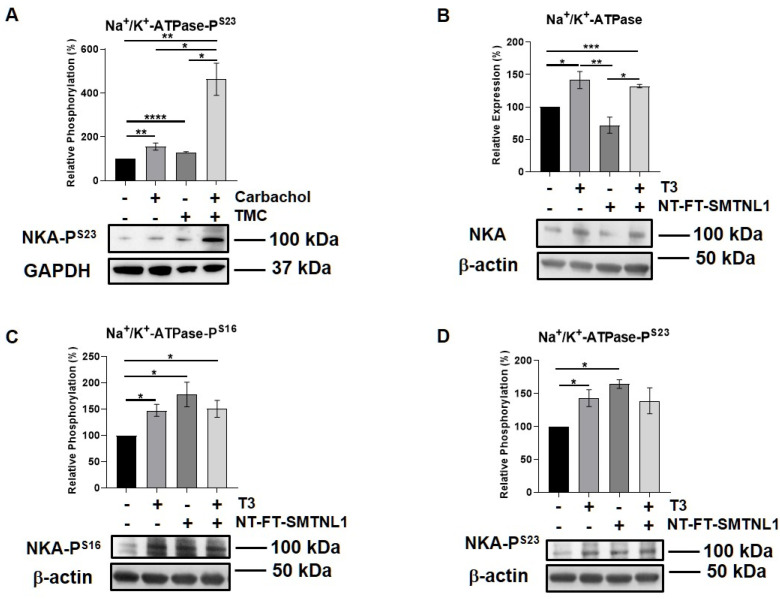
T_3_ Treatment and SMTNL1 overexpression influence the expression and phosphorylation of Na^+^/K^+^-ATPase in myotubes. (**A**) Plantaris muscle strips of mice were treated with 10 µM TMC for 30 min followed by 10 µM carbachol for 10 min. Samples were lysed and subjected to Western blot using anti-Na^+^/K^+^-ATPase. (**B**–**D**) Myoblasts were transfected with empty vector or NT-FT-SMTNL1 and were differentiated in parallel with a 72-h T_3_ treatment started from Day 4. Proteins from whole-cell lysates were separated by size and were analyzed using Western blot with anti-Na^+^/K^+^-ATPase (**B**), anti-Na^+^/K^+^-ATPase-P^S16^ (**C**) and anti-Na^+^/K^+^-ATPase-P^S23^ (**D**). Values represent *n* = 3, mean ± SEM. Data were normalized to the empty vector-transfected control. Differences between group means were determined using One-way ANOVA and Tukey’s post hoc test, *p* < 0.05 (*), *p* < 0.01 (**), *p* < 0.001 (***), *p* < 0.0001 (****).

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
