# Peer review of "Smoothelin-Like Protein 1 Regulates the Thyroid Hormone-Induced Homeostasis and Remodeling of C2C12 Cells via the Modulation of Myosin Phosphatase"

_ijms, 2021, doi:10.3390/ijms221910293_

Round 1

Reviewer 1 Report

  1. The author needs to clarify why choose 2 μg of pM13-NT-FT-SMTNL1 and cells received 10 nM of T3? And Does PM13-NT-FT-SMTNL1 have a dose effect?
  2. The results demonstrate that desmin levels were significantly elevated in the NT-FT-SMTNL1-transfected cells compared to the empty vector-transfected control on Day3 (Fig. 1A). Therefore, the author had better provide D3 immunofluorescence staining in Figure 1c.
  3. The author needs to clarify why different internal controls were selected for WB, especially for a-actin, b-actin, and a-tubulin in Fig. 3, And GAPDH and b-actin in Fig. 7.
  4. For WB, the added samples are not quite equal, especially in Figure 2. And If the author can also provide Q-RT data to coordinate with WB data, their results will be further confirmed.
  5. In Figure 5A, SMTNL1 T3 scratch image does not match the data of 5B and 5C.

Author Response

International Journal of Molecular Sciences- ijms-1361337

Response to Reviewers

We would like to thank the Reviewers for their constructive and helpful comments on our manuscript which greatly assisted us in the process of revision of the manuscript entitled "Smoothelin-like protein 1 Regulates the Thyroid Hormone-induced
Homeostasis and Remodeling of C2C12 Cells via the Modulation of Myosin
Phosphatase
".

 Response to Reviewer #1:

Comment 1.The author needs to clarify why choose 2 μg of pM13-NT-FT-SMTNL1 and cells received 10 nM of T3? And Does PM13-NT-FT-SMTNL1 have a dose effect?

Response 1.

We have conducted dose-dependent experiments on C2C12 cells with N-terminal Flag-tagged SMTNL1 (NT-FT-SMTNL1) vectors. As it is visible in the graph, the ratio of 2 mg DNA and 6 ml of transfection agent seemed to get the best efficiency.

Since we used in our experiments a well-characterized rodent muscle cell line (C2C12), we introduced to the cell culture experiments the concentrations measured in hyperthyroid mice. The plasma triiodothyronine, T3 concentration in hyperthyroid animals were 4.5±0.5*nM compared to 1.1±0.1 nM and hyperthyroid animals 0.1±0.06 nM (Branvold, D. J. et al, 2008; Thorson, L., 2014). This concentration is considered to be “supraphysiological” in several publications using C2C12 cell lines as hyperthyroid model (Yamauchi, M. et al, 2008). Moreover, Kenessey et al measured the TRalpha1 transcriptional activity using transient transfection of a thyroid hormone-responsive element (TRE) reporter plasmid. They found that T3 dose dependent and inversely proportional to nuclear TRalpha1 content, with 10 nM T3 having maximum effect (Kenessey,  et al, 2004).

Comment 2.The results demonstrate that desmin levels were significantly elevated in the NT-FT-SMTNL1-transfected cells compared to the empty vector-transfected control on Day3 (Fig. 1A). Therefore, the author had better provide D3 immunofluorescence staining in Figure 1c.

Response 2.

We have replaced the original image with a more representative set that includes the immunofluorescence staining of D0, D3 and D5 of empty-vector-transfected and NT-FT-SMTNL1-transfected cells. We have also modified the manuscript.

Comment 3.The author needs to clarify why different internal controls were selected for WB, especially for a-actin, b-actin, and a-tubulin in Fig. 3, And GAPDH and b-actin in Fig. 7.

Response 3.

The main criteria in the selection of loading controls were their independent expression both T3 and SMTNL1. We have the list of genes regulated by SMTNL1 as a transcription cofactor (Lontay et al, 2015, JBC) and none of the loading proteins were among them. We have also chosen b-actin, a-actinin and b-tubulin because their expression is not regulated by thyroid hormones. However, we have avoided using a-actin as internal control, since it is upregulated by T3 [Collie et al, Cell Growth Differ. 1992 Jan;3(1):31-42.]. The antibody that we used only recognizes the b isoform of actin. Loading controls were selected based on the membrane fragment available in each experiments since for semi-quantitative Western blot analysis we always applied the antibodies against the loading control on the same membrane as the protein of interests.  

Comment 4.For WB, the added samples are not quite equal, especially in Figure 2. And If the author can also provide Q-RT data to coordinate with WB data, their results will be further confirmed.

Response 4. 30 µg of proteins was loaded from each sample with a Hamilton syringe. However, the accuracy and precision of Hamilton syringes are known to be less than 100% causing small inequalities in the added samples. In order to eliminate these errors, proteins were normalized to their loading control, which is a widely used method for fixing this problem. There are many factors that can influence the half-life of mRNAs which have to go through many stages to become a fully mature, functional protein that is able to participate in signaling pathways. We were determined to investigate changes at the protein level since we tried to avoid the possibility of the posttranscriptional regulation of the proteins of interest in our model. Moreover, for the quantification of posstranslational modifications, the protein levels of the modified proteins are necessary to be assessed.

Comment 5.In Figure 5A, SMTNL1 T3 scratch image does not match the data of 5B and 5C. Response 5. Thank you for your comment. We have replaced the scratch image with a more representative one.

Reviewer 2 Report

In this study, the authors investigated the effects of supraphysiological vs. the physiological T3 levels on the expression and regulation of myosin phosphatase (MP) via the modulation of SMTNL1 in myoblast differentiation and the homeostasis of C2C12 myoblasts and myotubes. The authors concluded the differential regulation and role of MP in myoblasts and myotubes upon supraphysiological T3 exposure.

Comments

The reviewer has some concerns as follows:

  1. The data presentation and explanation for some results of this manuscript are inappropriate and a bit overstatement.
  2. There are no data for the protein expression of SMTNL1 in myoblasts or myotubes with or without T3 treatment. Moreover, the protein expression of SMTNL1 should also be confirmed in NT-FT-SMTNL1-treated myoblasts or myotubes.
  3. In the Introduction section, the authors mentioned “The objective of our work was to investigate the effect of supraphysiological vs. the physiological T3 level on the expression and regulation of MP…”. However, there is only one T3 concentration (10 nM) used throughout. Where are the supraphysiological and the physiological T3 levels happened?
  4. In Figures 1A and 1B, the statistical analysis is confusing and unconvincing. In empty vector group, there are no statistically significant differences among D0-D5 for desmin and MyHC. In SMTNL1 overexpression group, only D3 for desmin and only D5 for MyHC exhibit statistically significant difference. Moreover, in D5 for desmin and MyHC groups, there are no standard error bars shown in figures.
  5. In Figure 1C, the effects of SMTNL1 overexpression on desmin and phalloidin fluorescent signals are also not convincing. The changes in desmin fluorescent signals at D5 of SMTNL1 overexpression group are not pronounced. Moreover, why the phalloidin fluorescent signals are decreased compared to empty vector group? Are the actin filaments inhibited?
  6. The myogenesis for myotube formation in both empty vector control and SMTNL1 overexpression groups should be shown, including the morphology and myotube number counting.
  7. In Figures 2 and 3, the labels for protein expression in immunoblots are not correct, for example, in Fig. 2A, MCT8, but not vehicle or T3, was labeled. Moreover, why there are no standard error bars shown in D6-vehicle groups of these figures?
  8. In Figure 7, the experimental design for plantaris muscle strips of mice is confusing. These results from mouse muscle strips cannot be consistent with the in vitro findings. How about the effects of T3 or SMTNL1 overexpression on the function or structure of muscle strips?
  9. The authors should carefully check and revise the used internal control for Western blot analysis. Different figures need their internal control individually.

Author Response

International Journal of Molecular Sciences- ijms-1361337

Response to Reviewers

We would like to thank the Reviewers for their constructive and helpful comments on our manuscript which greatly assisted us in the process of revision of the manuscript entitled "Smoothelin-like protein 1 Regulates the Thyroid Hormone-induced
Homeostasis and Remodeling of C2C12 Cells via the Modulation of Myosin
Phosphatase
".

 Response to Reviewer #2:

Comment 1.The data presentation and explanation for some results of this manuscript are inappropriate and a bit overstatement.

Response 2. We have modified the manuscript by taking under consideration the comments in Comment 1.

Comment 2.There are no data for the protein expression of SMTNL1 in myoblasts or myotubes with or without T3 treatment. Moreover, the protein expression of SMTNL1 should also be confirmed in NT-FT-SMTNL1-treated myoblasts or myotubes.

Response 2. There are no data on SMTNL1 expression in myoblasts or myotubes upon T3 treatment, indeed, because we are going to publish these data in a recently accepted article [Major et al, 2021]. As you can see below, SMTNL1 expression significantly increases in response to T3 treatment (10 nM, 24 hours) in myoblasts. However, SMTNL1 expression significantly decreases in myotubes as a result of prolonged T3 treatment (10 nM, 72 hours).

We have successfully confirmed the overexpression of SMTNL1 protein using anti-Flag and anti-SMTNL1 antibodies in both myoblasts and myotubes. As indicated by the attached figures, SMTNL1 expression was increased by 85% in myoblasts, and by 43% in myotubes as a result of transfection. These figures have been incorporated into the Supplementary Material as Figure S2 and have modified the manuscript accordingly.

Comment 3.In the Introduction section, the authors mentioned “The objective of our work was to investigate the effect of supraphysiological vs. the physiological T3 level on the expression and regulation of MP…”. However, there is only one T3 concentration (10 nM) used throughout. Where are the supraphysiological and the physiological T3 levels happened?

Response 3.

We used in our experiments a well-characterized rodent muscle cell line (C2C12) and we introduced to the cell culture experiments the concentrations measured in hyperthyroid mice. The plasma triiodothyronine, T3 concentration in hyperthyroid animals were 4.5±0.5*nM compared to 1.1±0.1 nM and hyperthyroid animals 0.1±0.06 nM (Branvold, D. J. et al, 2008; Thorson, L., 2014). This concentration is considered to be “supraphysiological” in several publications using C2C12 cell lines as hyperthyroid model (Yamauchi, M. et al, 2008). Moreover, Kenessey et al measured the TRalpha1 transcriptional activity using transient transfection of a thyroid hormone-responsive element (TRE) reporter plasmid. They found that T3 dose dependent and inversely proportional to nuclear TRalpha1 content, with 10 nM T3 having maximum effect (Kenessey,  et al, 2004). 10nM is considered as supraphysiological and physiological is the normal concentration in the serum of the medium. The composition of FBS in the literature (see attached). According to this table, the FBS contains 1.2 ng/ml T3 and 0.12 ng/ml T4 in average. We thought that in case of such concentrations, there is no need to use striped medium because we wanted to do our experiments under physiological vs. supraphysiological conditions.

Comment 4.In Figures 1A and 1B, the statistical analysis is confusing and unconvincing. In empty vector group, there are no statistically significant differences among D0-D5 for desmin and MyHC. In SMTNL1 overexpression group, only D3 for desmin and only D5 for MyHC exhibit statistically significant difference. Moreover, in D5 for desmin and MyHC groups, there are no standard error bars shown in figures.

Response 4.

To prove the possible effect of SMTNL1 on myogenesis, we demonstrated the expression of early and late myogenic markers. To show the differences between the vehicle and SMTNL1-overexpressed groups in the course of myogenesis, differences between group means were determined by two-way ANOVA followed by Sidak multiple comparisons post hoc test (Figure 1A.,  p <0.05 (*), p <0.001 (***)). Since our research question was the effect of SMTNL1 and not the  differentiation process (which has been already investigated widely) we applied the above mentioned statictical analysis. To provide further information to represent that both desmin and MyH marekers are significantly altered during differentiation, we provided detailed graphical explanation of the differentiation. Our presentation shows that the rate of differentiation is not changing significantly after Day 4 since no significant differences either in MyHC or desmin expression level can be observed between Day 4 and 5. SMTNL1 accelerates differentiation since desmin expression shows significant change already at Day 3, while it in vehicle control groups at Day 4 compared to the Day 0 control.  However, MyHC late myogenic marker significant changes ae observable on at Day 4 on SMTNL1 overexpressed cells, a day later compared to the vehicle control, but it shows gradual significant changes all through differentiation al both cases.

Comment 5.In Figure 1C, the effects of SMTNL1 overexpression on desmin and phalloidin fluorescent signals are also not convincing. The changes in desmin fluorescent signals at D5 of SMTNL1 overexpression group are not pronounced. Moreover, why the phalloidin fluorescent signals are decreased compared to empty vector group? Are the actin filaments inhibited?

Response 5. We have replaced the original image with a more representative set that includes the immunofluorescence staining of D0, D3 and D5 of empty-vector-transfected and NT-FT-SMTNL1-transfected cells. We have also modified the manuscript. The overall expression level and actin content does not change either by SMTNL1 overexpression or by T3 exposure. The cytoskeletal structure is changing to the the process of myogenesis and the amount of filamentous actin is formed.

Comment 6.The myogenesis for myotube formation in both empty vector control and SMTNL1 overexpression groups should be shown, including the morphology and myotube number counting.

Response 6. Using light microscopy images with the same parameter settings, average area, average perimeter and myotube number were determined by ImageJ software for representative myoblasts and myotubes upon transfection with empty vector (MOCK) or recombinant SMTNL1 protein on Days 0, 3 and 5 of differentiation. All values were normalized to the average value of the control (day 0 of mock), due to single cells were the major components here. We observed that SMTNL1 expression significantly decreases the average area and perimeter of myotubes on Day 3 and Day 5 of differentiation, respectively. However, SMTNL1 overexpression markedly increased the number of myotubes on Day 3 and Day 5 of differentiation. Taken together, these data indicate that SMTNL1 overexpression leads to accelerated myogenesis with myotubes that are significantly smaller in size but bigger in number. We have incorporated these figures to the Supplementary Material (see Figure S2 inserted below).

Comment 7. In Figures 2 and 3, the labels for protein expression in immunoblots are not correct, for example, in Fig. 2A, MCT8, but not vehicle or T3, was labeled. Moreover, why there are no standard error bars shown in D6-vehicle groups of these figures?

Response 7. Both in Figure 2 and 3 the labels showed both vehicle and T3 (left side of the blots in each case). We have reanalyzed our data and presented the standard error in each groups including D6 vehicles. See modifications in the manuscript.

Comment 8.In Figure 7, the experimental design for plantaris muscle strips of mice is confusing. These results from mouse muscle strips cannot be consistent with the in vitro findings. How about the effects of T3 or SMTNL1 overexpression on the function or structure of muscle strips?

Response 8. The effect of SMTNL1 has been investigated in the skeletal muscle of WT and KO SMTNL1 mice (Lontay et al, 2015, Lontay et al, 2010). A primary target gene of SMTNL1 in these

muscles is myosin phosphatase-targeting subunit 1 (MYPT1). Deletion of SMTNL1 increases expression of MYPT1 30–40-fold in neonates and during development expression of both SMTNL1 and MYPT1 increases over 20-fold. As a reverse effect, the overexpression of SMTNL1 in our model resulted in a significant decrease in MYPT1 expression, and the activity of myosin phosphatase decreased resulted in an increase in the phosphorylation of Na+/K+ ATPase (NKA). The effect of TMC, the protein phosphatase 1 inhibitor resembles that of the SMTNL1 overexpression causing an increased phosphorylation of NKA.

Comment 9.The authors should carefully check and revise the used internal control for Western blot analysis. Different figures need their internal control individually.

Response 9. Thank you for your comment. We have replaced the images of the used internal controls where it is needed (Fig.1, Fig.6 and Fig.7).

Round 2

Reviewer 1 Report

The revised edition almost satisfied the reviewer's opinion and agreed to be published.

Reviewer 2 Report

The responses for reviewer's comments can be accepted. No further comments.